# FastSESR: Fast Scene-level Explicit Surface Reconstruction

**Jueqi Liu** [* 1 2]  **Xuechao Zou** [* 1 2]  **Congyan Lang** [1 2]

## Abstract

Explicit surface reconstruction aims to recover high-fidelity meshes directly from point clouds. While existing methods achieve strong performance on scene-level data, they often rely on test-time optimization, resulting in a prohibitive runtime of several minutes. To address this bottleneck, we propose FastSESR, a two-stage framework for efficient scene-level explicit surface reconstruction. In the first stage, a lightweight triangular candidate network (TCN) captures local connections via an edge-factorized parameterization, enabling effective extraction of surface triangles from uniformly sampled points. In the second stage, an offset optimization network amortizes offset refinement into a small, fixed number of learnable update steps guided by TCN, producing geometries that are more suitable for triangulation. Experiments on multiple scene-level datasets show that FastSESR accelerates surface reconstruction by at least $20\times$ over prior methods while maintaining competitive reconstruction quality. Moreover, evaluations on shape-level benchmarks indicate good generalization performance. Our code is available at https://github.com/Vaiduryasses/FastSESR.

## 1. Introduction

Surface reconstruction from point clouds is a foundational problem in computer vision and graphics, serving as a precondition for applications such as robotic navigation, autonomous driving, and digital content creation (Berger et al., 2017). Over the past decades, the field has evolved from classical combinatorial approaches, e.g., Delaunay trian-

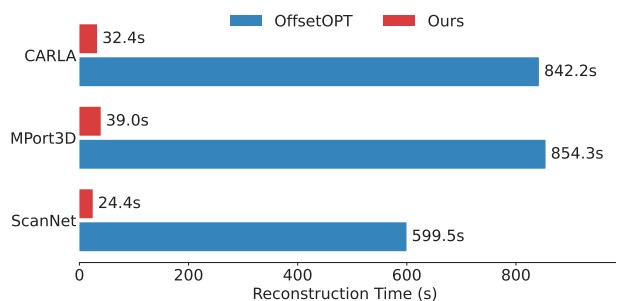

*Figure 1.* Reconstruction time comparison. We compare the reconstruction time of our method and the current SOTA method OffsetOPT (Lei, 2025) on scene-level datasets, and our method achieves a speedup of at least $20\times$.

gulation and Poisson-based reconstruction (Kazhdan et al., 2006; Kazhdan & Hoppe, 2013), to modern learning-based paradigms that represent geometry implicitly or explicitly. In recent years, implicit neural representations (Park et al., 2019; Mescheder et al., 2019; Ma et al., 2020; Gropp et al., 2020) have dominated the field by modelling geometry as continuous Signed Distance Fields (SDFs) or occupancy fields. However, these methods generally necessitate expensive post-processing steps (e.g., Marching Cubes (Lorensen & Cline, 1987) and its variants (Ju et al., 2002)) to extract explicit meshes, often struggling to recover high-frequency geometric details (Fan & Musialski, 2024). On the other hand, while scalable approaches like VDBFusion (Vizzo et al., 2022) and NKSR (Huang et al., 2023) enable the surface reconstruction of large-scale scenes via efficient volumetric integration or learned kernels, they fundamentally rely on sensor poses or high-quality oriented normals as inputs. Obtaining accurate normals from raw point clouds in the wild is often computationally intractable and error-prone, severely limiting the applicability of these pipelines.

Although learning-based explicit surface reconstruction methods (Lei et al., 2023; Liu et al., 2020; Rakotosaona et al., 2021; Sharp & Ovsjanikov, 2020) have received less attention than implicit counterparts, they usually can generate meshes directly from point clouds without requiring reliable normal estimation. Early learning-based approaches often suffer from poor edge-manifold quality, artifacts, and sensitivity to input sampling patterns (e.g., Poisson-disk sampling (Yuksel, 2015)). While

---

[*]Equal contribution  [1]School of Computer Science & Technology, Beijing Jiaotong University, Beijing, China [2]Key Laboratory of Big Data & Artificial Intelligence in Transportation, Ministry of Education, Beijing, China. Correspondence to: Congyan Lang <cylang@bjtu.edu.cn>.

*Proceedings of the $43^{rd}$ International Conference on Machine Learning*, Seoul, South Korea. PMLR 306, 2026. Copyright 2026 by the author(s).

optimization-based frameworks like OffsetOPT (Lei, 2025) and Point2Mesh (Hanocka et al., 2020) yield superior surface fidelity, their reliance on per-instance test-time optimization (TTO) necessitates hundreds of gradient steps, leading to prohibitively high computational latency.

To address the prohibitive latency of TTO while preserving high-fidelity reconstruction quality, we propose Fast-SESR, a two-stage network framework tailored for scene-level explicit surface reconstruction. Instead of performing per-instance iterative offset refinement, FastSESR produces per-scene geometry suitable for triangulation in a single feed-forward inference, reducing reconstruction time from minutes to seconds. In the first stage, a lightweight triangle candidate network decomposes the likelihood of triangle formation into edge-existence probabilities within local neighborhoods, enabling the reliable extraction of surface faces from uniform point clouds. In the second stage, an offset optimization network learns to refine point positions by predicting per-point offsets in an unsupervised manner, leveraging the connectivity cues provided by the triangle candidate network to move points toward configurations that are easier to triangulate. Extensive experiments across multiple scene-level datasets demonstrate that FastSESR achieves at least $20\times$ speedup over TTO methods while maintaining competitive reconstruction quality and generalizes favorably to shape-level benchmarks. In summary, our main contributions are as follows:

- We propose FastSESR, a two-stage framework mainly for scene-level explicit surface reconstruction. It discards the time-consuming test-time optimization paradigm and generates the surface with a single forward pass, achieving substantial improvement in inference speed from minutes to seconds, while maintaining high geometric quality. It also demonstrates good generalization performance on datasets at the shape level.
- We introduce a lightweight triangle candidate network with an edge-factorized parameterization to model the local patch topology. This approach decomposes the quadratic complexity of potential connections into compact unary terms and low-rank pairwise components, providing a minimal sufficient representation for estimation of triangle formation probabilities.
- We design an offset optimization network that amortizes test-time optimization into a small, fixed number of learnable refinement steps, guided by triangle-candidate signals from the triangle candidate network, achieving comparable quality while removing hundreds of steps of iterative optimization at inference.

## 2. Related Work

### 2.1. Implicit Surface Reconstruction

Implicit neural representations, such as DeepSDF (Park et al., 2019) and Occupancy Networks (Mescheder et al., 2019), model geometry as continuous signed distance or occupancy fields, having been extended to open surfaces by unsigned formulations (e.g., NDFs) (Chibane et al., 2020). To improve efficiency and scalability in practical applications, volumetric fusion systems based on sparse data structures (e.g., TSDF/VDB) have been explored (Vizzo et al., 2022), and neural-kernel implicit reconstruction approaches such as NKSR reconstruct implicit surfaces from large, sparse point clouds with strong inductive biases and memory-efficient solvers (Huang et al., 2023). However, these high-efficiency methods still rely on accurate poses (Vizzo et al., 2022) or oriented normals for surface reconstruction (Huang et al., 2023), which can be unreliable for raw point clouds. Moreover, implicit representations may require post-processing to extract the mesh and can oversmooth sharp geometric details (Fan & Musialski, 2024).

### 2.2. Explicit Surface Reconstruction

Recent learning-based explicit methods aim to recover triangle meshes directly from unoriented point clouds, bypassing the need for reliable normal estimation. Methods such as PointTriNet (Sharp & Ovsjanikov, 2020), DSE (Rakoto-saona et al., 2021), and CircNet (Lei et al., 2023) construct local surface elements or local connectivity primitives (e.g., candidate faces or circumcenters), but often suffer from artifacts (e.g., holes or noise) and are sensitive to input sampling distributions (Yuksel, 2015). Despite these feed-forward approaches, test-time optimization has also been explored to enforce geometric consistency via per-instance iterative refinement. For example, Point2Mesh (Hanocka et al., 2020) optimizes a per-shape CNN self-prior to shrink-wrap a deformable template, exhibiting stronger tolerance to noise and missing regions. Notably, most of the above methods mainly target shape-level surface reconstruction of single objects. More recently, OffsetOPT (Lei, 2025) advances this line by learning a local geometric prior and performing test-time refinement, achieving impressive geometric fidelity. However, its reliance on per-instance iterative optimization becomes a critical bottleneck when scaling to scene-level surface reconstruction: processing a large scene can take several minutes, making it impractical for real-time applications. In contrast, our approach uses a network (OON) to directly predict point-wise offsets, enabling per-scene triangle-mesh generation in a single forward pass.

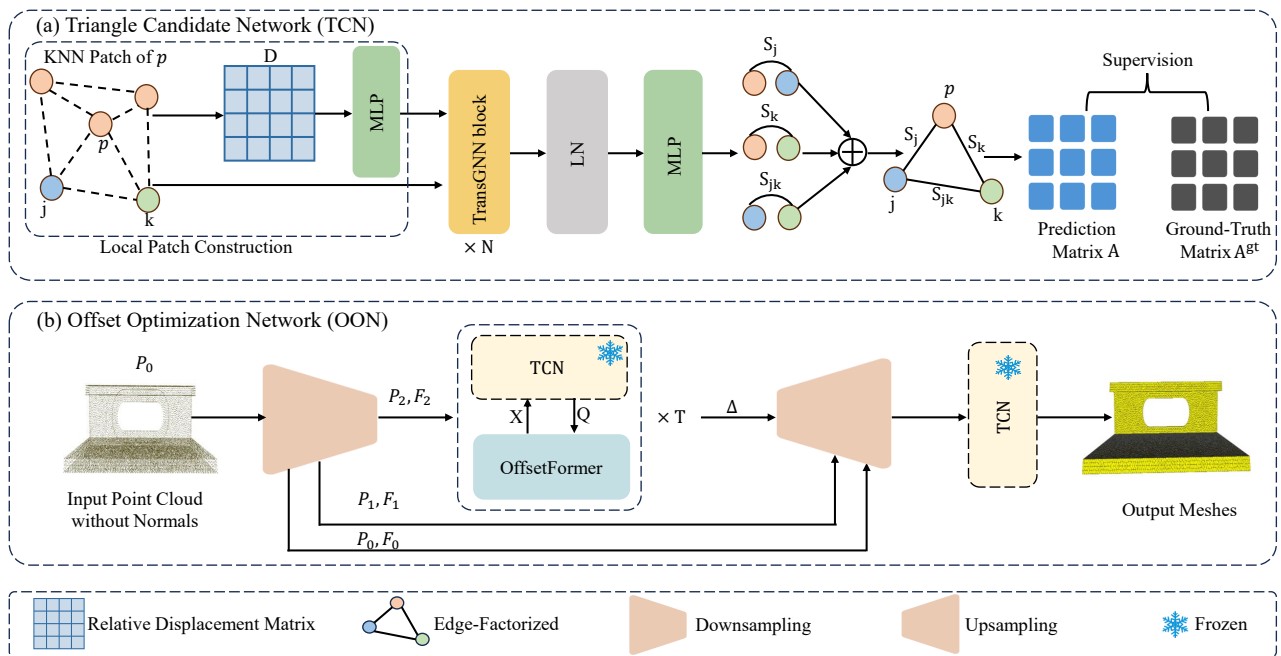

*Figure 2.* Overview of the proposed FastSESR. (a) TCN: Given each point's local $K$NN patch, TCN predicts a per-patch triangle-candidate probability matrix over neighbor pairs, providing compact connectivity priors for subsequent surface extraction and offset optimization. (b) OON: it learns offsets via TCN-guided attention on local patches, achieving fast feed-forward alignment with reasonable connections.

# 3. Method

As shown in Figure 2, our model comprises the triangle candidate network (TCN) and the offset optimization network (OON). In Section 3.1, we introduce TCN, which learns from uniform point clouds to predict the existence of local triangles. In Section 3.2, we explain how TCN guides OON to learn the offsets. In Section 3.3, we will delineate the supervised training loss of TCN and the unsupervised training loss of OON separately. During reconstruction, we directly use the trained OON to obtain the updated point cloud, then use the frozen TCN to extract the surface.

## 3.1. Triangle Candidate Network

**Local Patch Construction.** Following the conventions of prior works (Lei et al., 2023; Qi et al., 2017), we leverage local geometric relationships to construct features. We first employ $K$NN to capture the neighborhood of each point (Preparata & Shamos, 2012), obtaining $K$ nearest neighbors $\overline{\mathcal{K}}(p) = \{p_i\}_{i=1}^{K}$ of point $p$. The network computes the relative coordinates of each point with respect to the center to normalize the $K$NN neighborhood and serves as the initial geometric input:

$$\mathcal{K}(p) = \left\{ p_k \mid p_k = 0.01 \frac{p_k - p}{\|p_1 - p\|} \right\}_{k=1}^{K}. \quad (1)$$

where $q_k$ denotes the $k$-th nearest neighbor of $p$. The network then takes $\mathcal{K}(p)$ with positional encoding (Mildenhall et al., 2021) as feature $\mathbf{x} \in \mathbb{R}^{(K+1)\times C}$. To capture high-order geometric correlations, we further construct a relative displacement matrix $D \in \mathbb{R}^{(K+1)\times(K+1)\times 3}$, which is an antisymmetric matrix where each element represents the relative displacement between corresponding points in the local point cloud patch. Then, we map it to a relative position bias through an MLP to encode the spatial dependency between point pairs.

**TransGNN Feature Refinement.** Given features $\mathbf{x}$, the local patch $\mathcal{K}(p)$, and the relative position bias from $D$, a TransGNN block updates features via a GNN layer followed by self-attention. We adopt a pre-norm design (Xiong et al., 2020) with a geometry-conditioned bias $B \in \mathbb{R}^{(K+1)\times C}$ (obtained by projecting the relative position coordinates) to get input $\hat{\mathbf{x}}$. For each point $i$ in $\mathcal{K}(p)$, we first build a $K_{msg}$NN graph to compute the $K_{msg}$ nearest neighbors $\mathcal{N}(i)$ in the patch, then we pass the message on the $K_{msg}$ nearest neighbors ($K_{msg} \ll K$) in the GNN layer (Veličković et al., 2017):

$$\alpha_{ij} = \text{softmax}_{j \in \mathcal{N}(i)}\Big(\text{MLP}([\hat{\mathbf{x}}_i, \hat{\mathbf{x}}_j])\Big), \quad (2)$$

$$m_{ij} = \text{MLP}([\hat{\mathbf{x}}_i, \hat{\mathbf{x}}_j - \hat{\mathbf{x}}_i]), \quad (3)$$

$$y_i = \sum_{j \in \mathcal{N}(i)} \alpha_{ij} m_{ij}, \quad (4)$$

here $y \in \mathbb{R}^{(K+1) \times C}$, $\alpha_{ij}$ denotes the softmax-normalized attention weight measuring the importance of neighbor $j$ to node $i$, while $m_{ij}$ denotes the message vector generated by an MLP from the center feature and relative feature difference to be aggregated for updating node $i$. Next, we apply multi-head self-attention over the $K + 1$ tokens with relative bias injection (Vaswani et al., 2017):

$$\text{Attn}(y) = \text{softmax}\left(\frac{\mathbf{Q}\mathbf{K}^{\top}}{\sqrt{d}} + \text{MLP}(D)\right)\mathbf{V}, \quad (5)$$
$$[\mathbf{Q}, \mathbf{K}, \mathbf{V}] = y\mathbf{W}_{qkv}.$$

A lightweight MLP is finally used for feature recombination.

**Edge-Factorized Parameterization.** Directly regressing a dense triangle candidate matrix over neighbor pairs requires predicting $K^2$ logits per local patch (Lei, 2025), which is inefficient and lacks geometric interpretability. To address this, we introduce an edge-factorized parameterization that decomposes the neighbor-pair logit $S_{jk}$ into two unary terms and one pairwise term:

$$S_{jk} = s_j + s_k + s_{jk}, \qquad j, k \in \{1, \dots, K\}, \quad (6)$$

Here, $\mathbf{Y} = [\mathbf{y}_1, \dots, \mathbf{y}_K]^{\top} \in \mathbb{R}^{K \times C}$ denotes the neighbor token features output by the final TransGNN block (after layer normalization), where $\mathbf{y}_j \in \mathbb{R}^C$ is the feature of neighbor $j$. A lightweight MLP maps $\mathbf{Y}$ to a unary score vector $\mathbf{s} = [s_1, \dots, s_K]^{\top} \in \mathbb{R}^K$, where $s_j$ measures the individual compatibility of selecting neighbor $j$ with the center point. The pairwise term $s_{jk}$ models the interaction between neighbors $(j, k)$ and is computed from projected neighbor features:

$$s_{jk} = \langle \mathbf{u}_j, \mathbf{u}_k \rangle, \qquad \mathbf{u}_j = \phi(\mathbf{y}_j) \in \mathbb{R}^r, \quad (7)$$
$$j, k \in \{1, \dots, K\},$$

where $\phi(\cdot)$ is a bias-free linear projection applied to each neighbor token feature. This yields a compact score matrix

$$\mathbf{S} = \mathbf{s}\mathbf{1}^{\top} + \mathbf{1}\mathbf{s}^{\top} + \mathbf{U}\mathbf{U}^{\top}, \quad (8)$$

reducing the pairwise parameterization from predicting $K^2$ independent logits to computing $\mathbf{U} \in \mathbb{R}^{K \times r}$ with $r < K$ while preserving symmetry. We further apply a diagonal mask to invalidate $j = k$ pairs and symmetrize $\mathbf{S}$ to ensure consistent neighbor-pair scoring. Finally, the candidate probability matrix for the triangle is $\mathbf{A} = \sigma(\mathbf{S}) \in \mathbb{R}^{K \times K}$.

**Surface Extraction.** Following OffsetOPT (Lei, 2025), we sort the columns of each row in the predicted probability $A$ of each point to select the two triangles with the highest probabilities, thereby extracting the adjacent triangles of each point. At the same time, we set the confidence thresholds to 0.8 and 0.5 to filter the selected triangles to avoid selecting triangles with too low probabilities. We check the angle between the two triangles to avoid unnecessary face folding and enforce the angle to be greater than 120°. Repetitive triangle predictions from different points are deleted.

## 3.2. Offset Optimization Network

Since our triangle candidate network is trained on the shape-level dataset ABC (Koch et al., 2019) with uniform point clouds, it is highly challenging to generalize to large-scale scene datasets with diverse point cloud distributions, such as ScanNet (Dai et al., 2017). Therefore, we designed an offset optimization network to fine-tune the positions of the points. Specifically, we employed a Unet-style network (Ronneberger et al., 2015).

**Downsampling.** Given the inherent complexity of point cloud geometry, single-scale feature extraction often fails to balance local details with global topology. Therefore, OON employs a multi-scale architecture similar to Dynamic Graph CNNs (DGCNN)(Wang et al., 2019). For an input point cloud $P \in \mathbb{R}^{N \times 3}$, the network generates a hierarchy of point sets $\{P_l\}_{l=0}^L$ and corresponding feature maps $\{F_l\}_{l=0}^L$ through successive downsampling operations, where $F_l \in \mathbb{R}^{N_l \times C_l}$, $l = 0$ represents the original resolution and $l = L$ corresponds to the coarsest bottleneck level. Feature extraction at each resolution is performed using a cross-set EdgeConv (CS-EdgeConv) operator, inspired by PRSCN (Zhu et al., 2021). This operator enables both intra-scale and cross-scale geometric reasoning. Given a query point set $P_l = \{p_i\}_{i=1}^{N_l}$ and a different key set $P_{l-1} = \{p_j\}_{j=1}^{N_{l-1}}$, we build a $k$NN graph from $P_l$ to $P_{l-1}$ and compute the output feature for each query point $p_l$ as:

$$F_l = \max_{j \in \mathcal{N}_k(i)} h_\Theta\left(F_i^{(l)}, F_j^{(l-1)} - F_i^{(l)}\right), \quad (9)$$

where $\mathcal{N}_k(i)$ denotes the $k$ nearest neighbors of $p_i$ in the key set $P_{l-1}$, $F_i^{(l)}$ and $F_j^{(l-1)}$ are the query-set and key-set features respectively, and $h_\Theta$ is a learnable edge function implemented by an MLP. When $l = 0$, this operator naturally reduces to the standard EdgeConv. CS-EdgeConv formulation allows information to propagate across levels and equips the network with strong sensitivity to local topological variations in complex scenes.

**Shared OffsetFormer Update.** At the coarsest level $L$, we unroll refinement for a fixed number of $T$ steps and share the same OffsetFormer parameters across all steps. We initialize the offsets as

$$\Delta_0 = \mathbf{0} \in \mathbb{R}^{N_L \times 3}, P_{in} = P_L + \Delta_0, \quad (10)$$

so that the initial geometry equals the coarsest point set. At step $t$, the current geometry is

$$P_{t+1} = P_t + \Delta_{t+1}, \quad t \in \{0, \dots, T-1\}, \quad (11)$$

here $P_0 = P_{in}$, $\Delta_T = \Delta_L$. For point $i \in P_t$, we form a local patch using relative coordinates, like the first stage and feed them into the TCN, yielding connectivity logits over local candidate triangles:

$$S_{t,i} = TCN(\mathbf{x}_{t,i}), \quad (12)$$

let $A_{t,i} = \sigma(S_{t,i})$ be the corresponding probabilities. We summarize $S_{t,i}$ into a compact statistic vector

$$\mathcal{Q}_{t,i} = \Big[ \max(A_{t,i}), \ \mathrm{mean}(A_{t,i}), \ H(A_{t,i}), \ \mathrm{gap}(S_{t,i}) \Big], \tag{13}$$

where $H(A)$ is the averaged binary entropy, and $\mathrm{gap}(S)$ denotes the top-2 logit margin. Similar entropy-aware signals have been used for adaptive step alignment in diffusion policy optimization (Yan et al., 2025). In our framework, $Q_t$ serves as a compact geometry-aware confidence descriptor for TCN-guided offset refinement. OffsetFormer uses $\mathcal{Q}_t$ as an additional conditioning feature without handcrafted gating, fuses it with the current hidden state and bottleneck features, and performs geometry-conditioned neighborhood aggregation via vector attention with relative geometric encoding:

$$(\Delta_{t+1}, h_{t+1}) = \mathcal{G}_\theta(h_t, P_t, \mathcal{Q}_t, F_L), \tag{14}$$

where $\Delta_{t+1}$ is produced by the operator at each forward step, and its magnitude is regulated by a unified learnable step-size control to ensure stable fixed-step refinement. We initialize $h_0$ by feeding a per-point geometric descriptor formed from the nearest-neighbor distance $d_0$ (and $d_0^2$, padded with zeros to 6 dims) into a lightweight MLP. In particular, an OffsetFormer operator does the following:

(i) **Token Construction**: We build a per-point token representation by concatenating the current state:

$$z_t = \mathrm{Embedding}(\mathrm{cat}(h_t, P_t, \mathcal{Q}_t, F_L)). \tag{15}$$

(ii) **Vector attention** (Wu et al., 2022): We update tokens by aggregating neighbor messages on a $k$NN graph:

$$\hat{z}_t = \mathrm{VecAttn}(z_t, \mathcal{N}_t), \tag{16}$$

where $\mathcal{N}_t$ is the $k$NN neighborhood computed from the current point positions, and $\mathrm{VecAttn}(\cdot)$ denotes the vector attention module for message aggregation.

(iii) **Delta prediction**: Finally, the aggregated features pass through an MLP to update the hidden state and predict the position correction $\Delta_t$:

$$h_{t+1} = \mathrm{LayerNorm}(\hat{z}_t + \mathrm{MLP}(\hat{z}_t)), \tag{17}$$
$$\Delta_{t+1} = \gamma \cdot (h_{t+1} W_{\mathrm{out}} + b_{\mathrm{out}}), \tag{18}$$

where $W_{\mathrm{out}} \in \mathbb{R}^{d \times 3}$, $h_{t+1} \in \mathbb{R}^{N_L \times d}$, and $\gamma$ is a learnable scalar initialized to a small value (e.g., $10^{-4}$) to ensure stable optimization in early training stages.

**Upsampling.** After predicting coarse-level offsets $\Delta_L$ at the bottleneck, FastSESR recovers full-resolution offsets via a learnable coarse-to-fine upsampling. Specifically, when upsampling from level $l+1$ to $l$, we first obtain an initial estimate $\widetilde{\Delta}_l$ by Inverse Distance Weighting (IDW) (Shepard,

1968) from the coarser set $P_{l+1}$ to the denser set $P_l$. We then concatenate $\widetilde{\Delta}_l$ with the skip-connected features $F_l$ from the downsampling and predict a residual refinement using a lightweight MLP:

$$\widetilde{\Delta}_l = \mathrm{IDW}(\Delta_{l+1}; P_{l+1} \to P_l), \tag{19}$$
$$\delta_l = \mathrm{MLP}([\widetilde{\Delta}_l, \ F_l]), \tag{20}$$
$$\Delta_l = \widetilde{\Delta}_l + \delta_l. \tag{21}$$

This procedure is repeated until fine-grained offsets $\Delta_0$ are obtained. This hierarchical design keeps the core solver at the coarsest resolution for efficiency, while the learnable upsampling module transfers and refines offsets at higher resolutions without introducing additional optimization loops. In this way, FastSESR retains an iterative coarse-to-fine refinement behavior within a compact feed-forward pipeline.

### 3.3. Loss Function

For TCN, we use BCE loss between the predicted probabilities $A$ and the ground-truth labels $A^{gt}$, where the ground-truth labels $A^{gt}$ are generated from the adjacent triangles of each point in the training meshes (Lei, 2025). OON is trained fully unsupervised without ground-truth meshes. Given an input point cloud, we first downsample and obtain the coarsest-level points $P_L$. To make training feasible on large-scale scenes (e.g., ScanNet), we further subsample at this level using farthest point sampling (FPS) and map the selected indices back to the input, yielding a subset $P_0^{\mathrm{sub}}$. The offset network predicts $\Delta_{\mathrm{pred}}$ on this subset, producing the deformed points:

$$P^{\mathrm{sub}} = P_0^{\mathrm{sub}} + \Delta_{\mathrm{pred}}. \tag{22}$$

For datasets with limited point clouds (e.g., ABC), we still use all point clouds for training. For each deformed point, we build a local $K$NN patch within the subset and feed it into the frozen TCN to yield connectivity logits $S$. Following OffsetOPT (Lei, 2025), we generate pseudo labels via a row-wise Top-2 rule: for each row of $S$, the two largest logits are marked as positives (filtered by a confidence threshold $\tau = 0.5$), forming a binary mask $M$. Finally, the total learning objective is formulated as:

$$\mathcal{L} = \mathrm{BCEWithLogits}(S, M) + \lambda \left\| \Delta_{\mathrm{pred}} \right\|_2^2, \tag{23}$$

where the first term enforces connectivity consistency with the pseudo labels, and the second term is an $L_2$ regularizer weighted by $\lambda$ to prevent the predicted offsets $\Delta_{\mathrm{pred}}$ from excessive drifting during offsets update.

## 4. Experiment

### 4.1. Experimental Setup

**Datasets.** We use the ABC dataset (Koch et al., 2019) to train the TCN network. For the ABC dataset, which

*Table 1.* Quantitative comparison on scene-level datasets: ScanNet, Matterport3D, CARLA. * indicates the usage of ground-truth normals. "0.3M" denotes randomly sampling 300k points from the ground-truth mesh.

| | Method | Geometric Accuracy | | Surface Quality | | |
|---|---|---|---|---|---|---|
| | | CD1($\times 10^2$)↓ | CD2($\times 10^5$)↓ | F1↑ | NC↑ | NR↓ |
| ScanNet | SPSR* (Kazhdan & Hoppe, 2013) | 5.428 | 352.674 | 0.194 | 0.709 | 35.904 |
| | NKSR* (Huang et al., 2023) | 0.157 | 0.164 | 0.997 | 0.963 | 9.824 |
| | NKSR | 0.423 | 1.648 | 0.793 | 0.901 | 17.699 |
| | OffsetOPT (Lei, 2025) | **0.147** | **0.136** | **1.0** | 0.960 | 9.533 |
| | **FastSESR (Ours)** | 0.154 | 0.261 | 0.999 | **0.964** | **9.023** |
| MPort3D | SPSR* | 0.926 | 28.893 | 0.724 | 0.830 | 23.322 |
| | NKSR* | 0.183 | 0.220 | 0.995 | 0.936 | 12.713 |
| | NKSR | 0.271 | 0.762 | 0.939 | 0.894 | 18.076 |
| | CircNet (0.3M) (Lei et al., 2023) | 0.447 | 1.283 | 0.773 | 0.855 | 20.009 |
| | OffsetOPT | **0.148** | **0.139** | **1.0** | 0.938 | 10.665 |
| | **FastSESR (Ours)** | 0.159 | 0.171 | 0.997 | **0.940** | **10.543** |
| CARLA | SPSR* | 4.407 | 234.338 | 0.121 | 0.733 | 30.835 |
| | NKSR* | 0.175 | 0.299 | 0.974 | 0.953 | 7.740 |
| | NKSR | 0.238 | 2.682 | 0.968 | 0.936 | 10.761 |
| | OffsetOPT | 0.124 | 0.272 | 0.987 | 0.963 | 5.530 |
| | **FastSESR (Ours)** | **0.114** | **0.171** | **0.990** | **0.967** | **4.929** |

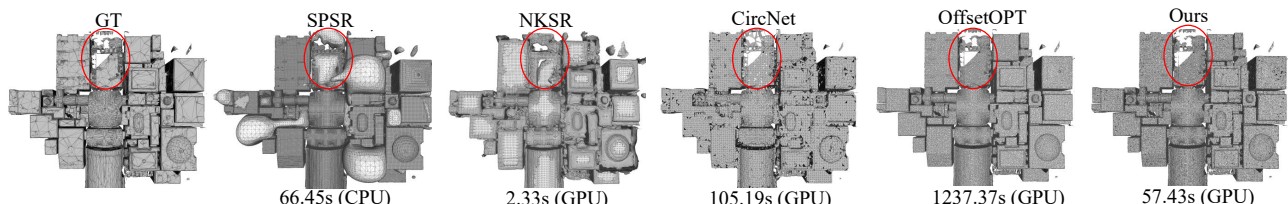

*Figure 3.* Visual comparison on Matterport3D. NKSR and SPSR introduce spurious geometry in the upper region, while CircNet shows substantial holes (black). Our method and OffsetOPT suppress these artifacts and produce higher-fidelity surfaces. Reconstruction times are reported below. Because SPSR is currently available only as a CPU implementation, we evaluate it exclusively on the CPU.

contains 9,026 voxelized meshes, we use 25% for training and the other 75% for testing, as in OffsetOPT (Lei, 2025). To verify the effectiveness of the offset optimization network, we use large-scale scene datasets ScanNet (Dai et al., 2017), Matterport3D (Chang et al., 2017), as well as CARLA (Dosovitskiy et al., 2017). Although our framework is mainly designed for large-scale datasets, we still conduct experiments on object-level shape datasets to verify its generalization ability. Training OON on the ABC dataset, we use FAUST (Bogo et al., 2014) and MGN (Bhatnagar et al., 2019) to generalize, which are respectively the human body dataset and the clothing dataset. More details about datasets can be found in Appendix A.

**Implementation Details.** We implement our framework on a single NVIDIA RTX 3090 GPU. Both of the networks are optimized using the Adam optimizer (Kingma & Ba, 2015) with an initial learning rate of $10^{-3}$ and weight decay of $10^{-4}$. $K$ is set to 49, and $k$ is 16. We employ five TransGNN blocks and two OffsetFormer modules for large-scale scene datasets, and use three OffsetFormer modules for object-level datasets. We perform two downsampling operations at a rate of $0.25$ each time and set the rank $r$ to 32. For the second stage, the regularization weight $\lambda$ is set to $10^{-3}$. Although ground-truth (GT) meshes are withheld during training, we utilize them during the validation phase for model selection. Specifically, we compute the bidirectional Chamfer Distance (CD) between the predicted geometry and the GT surface on the validation set, saving the checkpoint that minimizes this metric. Although our method can handle general point cloud inputs, our goal is to reconstruct surfaces from dense point clouds. Moreover, the scale of large-scale scene datasets varies greatly. Therefore, similar to general surface reconstruction methods (Huang et al., 2023; Kazhdan & Hoppe, 2013; Lei, 2025), we still voxelize the dense point clouds of large-scale scene datasets to create a more regular input (using the maximum nearest-neighbor distance as the scale), and then apply our two-stage network to reconstruct the surface.

*Table 2.* Quantitative comparison on shape-level datasets (ABC, FAUST, and MGN). We verify the generalization ability of our FastSESR against state-of-the-art methods. $^{\dagger}$ and * denote results using estimated normals and ground-truth normals, respectively. "0.1M" indicates randomly sampling 100k points from the ground-truth mesh.

| | Method | Geometric Accuracy | | Surface Quality | | |
|---|---|---|---|---|---|---|
| | | CD1($\times 10^2$)↓ | CD2($\times 10^5$)↓ | F1↑ | NC↑ | NR↓ |
| **ABC** | ball-pivot$^{\dagger}$ (Bernardini et al., 1999) | 0.297 | 0.684 | 0.939 | 0.981 | 2.244 |
| | SPSR* (Kazhdan & Hoppe, 2013) | 0.400 | 6.081 | 0.901 | 0.972 | 6.020 |
| | DSE (Rakotosaona et al., 2021) | 0.285 | 0.548 | 0.949 | 0.985 | 1.793 |
| | PointTriNet (Sharp & Ovsjanikov, 2020) | 0.288 | 0.790 | 0.948 | 0.984 | 1.931 |
| | CircNet (Lei et al., 2023) | 0.284 | 0.544 | 0.950 | 0.985 | 1.758 |
| | NKSR* (0.1M) (Huang et al., 2023) | 0.306 | 1.167 | 0.938 | **0.990** | 2.929 |
| | OffsetOPT (Lei, 2025) | 0.283 | 0.540 | 0.951 | 0.988 | 1.318 |
| | **FastSESR (Ours)** | **0.283** | **0.540** | **0.951** | 0.988 | **1.311** |
| **FAUST** | ball-pivot$^{\dagger}$ | 0.323 | 1.002 | 0.923 | 0.970 | 6.037 |
| | SPSR* | 0.427 | 4.108 | 0.915 | 0.969 | 10.269 |
| | DSE | 0.218 | 0.307 | 0.995 | 0.984 | 3.910 |
| | PointTriNet | 0.219 | 0.308 | 0.995 | 0.983 | 4.393 |
| | CircNet | 0.221 | 0.316 | 0.993 | 0.980 | 4.557 |
| | NKSR* (0.1M) | 0.227 | 0.319 | **0.997** | 0.987 | 6.303 |
| | OffsetOPT | **0.217** | **0.301** | 0.996 | 0.985 | 4.038 |
| | **FastSESR (Ours)** | 0.222 | 0.328 | 0.991 | **0.987** | **3.727** |
| **MGN** | ball-pivot$^{\dagger}$ | 0.462 | 4.917 | 0.844 | 0.974 | 5.803 |
| | SPSR* | 1.077 | 10.481 | 0.402 | 0.948 | 12.224 |
| | DSE | 0.270 | 0.530 | **0.968** | 0.983 | 3.970 |
| | PointTriNet | 0.272 | 0.562 | 0.967 | 0.981 | 4.398 |
| | CircNet | **0.269** | 0.512 | **0.968** | 0.981 | 4.230 |
| | NKSR* (0.1M) | 0.381 | 0.884 | 0.891 | 0.990 | 4.997 |
| | OffsetOPT | 0.278 | **0.511** | 0.964 | 0.991 | 2.967 |
| | **FastSESR (Ours)** | 0.284 | 0.574 | 0.957 | **0.992** | **2.687** |

*Table 3.* Inference time comparison on large-scale datasets between OffsetOPT and FastSESR. We report the average time per scene. The speedup is calculated as $T_{\text{OffsetOPT}}/T_{\text{FastSESR}}$.

| Dataset | OffsetOPT(s) | FastSESR(s) | Speedup |
|---|---|---|---|
| ScanNet | 599.47 | **24.43** | **24.52×** |
| Matterport3D | 854.31 | **39.04** | **21.88×** |
| CARLA | 842.16 | **32.36** | **26.02×** |

## 4.2. Scene and Shape Surface Reconstruction

**Methods Comparison.** We compare FastSESR against explicit surface reconstruction methods: Ball-Pivoting Algorithm (BPA) (Bernardini et al., 1999), DSE (Rakotosaona et al., 2021), PointTriNet (Sharp & Ovsjanikov, 2020), CircNet (Lei et al., 2023), and OffsetOPT (Lei, 2025), as well as implicit methods NKSR (with normals) (Huang et al., 2023) and SPSR (with normals) (Kazhdan & Hoppe, 2013).

**Evaluation Metrics.** We evaluate geometric fidelity using five standard metrics: Chamfer Distance (CD1, CD2), F-

Score (F1), Normal Consistency (NC), and Normal Reconstruction Error (NR) in Degrees, following OffsetOPT (Lei, 2025) and CircNet (Lei et al., 2023). To quantify the efficiency advantage of our network, we record the average inference time per sample for large-scale datasets. More details of these metrics can be seen in Appendix B.

**Performance Comparison on Scene Datasets.** Table 1 shows the results between different methods. For scene-level surface reconstruction, input points are voxelized with grid sizes of 2 cm for ScanNet and 10 cm for Matterport3D and CARLA. On scene-level benchmarks, FastSESR substantially improves surface quality, achieving higher normal consistency (NC) and lower normal reconstruction error (NR), while maintaining competitive geometric accuracy as measured by Chamfer Distance (CD1/CD2) and F-score. On CARLA, which features dense point clouds and complex, large-scale layouts, our method achieves the best overall performance among the compared methods. Figure 3 presents qualitative comparisons on Matterport3D: NKSR and SPSR tend to generate spurious geometry in regions without un-

derlying surfaces, whereas CircNet produces more holes (black regions). In contrast, our method and OffsetOPT deliver cleaner reconstructions with better detail preservation. Meanwhile, the reconstruction times reported below also indicate that, among all three explicit surface reconstruction methods, our method achieves the shortest runtime.

**Efficiency Comparison on Scene Datasets.** We reported reconstruction latency in Table 3 and Figure 1. OffsetOPT (Lei, 2025) relies on per-instance test-time optimization with iterative gradient descent, leading to minutes-level runtime for a single large scene. In contrast, FastSESR amortizes the iterative offset search into a fixed number of weight-shared refinement steps, enabling low-latency inference: it reconstructs large scenes within tens of seconds and completes object surface reconstruction in a single compact forward pass. Overall, FastSESR achieves at least $20\times$ speedup over test-time optimization baselines while retaining the benefits of iterative refinement.

**Generalization Comparison on Shape Datasets.** Note that on the object-level shape dataset, since TCN has already been trained on the ABC dataset, for the offset optimization network OON we only perform one epoch of training, and then directly use this model to generalize to FAUST and MGN datasets. As shown in Table 2, our approach, similar to large-scale datasets, remains close to the current state of the art in geometric accuracy while achieving good surface quality. More visualizations are available in Appendix C.

### 4.3. Ablation Studies

*Table 4.* Quantitative comparison of geometric quality and efficiency between learnable offset optimization and test-time optimization on CARLA.

| Method | CD1($\times10^2$)↓ | CD2($\times10^5$)↓ | F1↑ | NC↑ | NR↓ | Time(s) |
|---|---|---|---|---|---|---|
| TTO ($T=2$) | 6.745 | 359.334 | 0.048 | 0.666 | 41.959 | 252.65 |
| TTO ($T=5$) | 6.740 | 358.025 | 0.047 | 0.663 | 42.210 | 259.40 |
| TTO ($T=10$) | 6.715 | 354.972 | 0.049 | 0.663 | 42.258 | 269.94 |
| Learnable | **0.114** | **0.171** | **0.990** | **0.967** | **4.929** | **32.36** |

**Learnable Update vs. Test-Time Optimization.** To mechanistically validate the capacity of learnable updates, we designed a hybrid experiment: we froze the parameters of the trained downsampling and upsampling in FastSESR, but replaced the intermediate OffsetFormer operator with an SGD-based iterative test-time optimization process that optimizes offsets in the feature space. Results in Table 4 demonstrate that after multiple rounds of iteration, the SGD-based iterative optimization is inferior to our FastSESR in all aspects. It supports the claim that OffsetFormer has learned a more efficient update rule than general test-time optimization, which can achieve better geometric results within a fixed step-count budget. More ablation studies about OffsetFormer can be found in Appendix D.

*Table 5.* Parameter comparison between dense pairwise prediction and our edge-factorized design.

| Method | Params (M) ↓ |
|---|---|
| Dense Pairwise Prediction | 6.08 |
| Edge-Factorized | **0.35** |

*Table 6.* Quantitative comparison between dense pairwise prediction and our edge-factorized design on the ABC dataset.

| Method | CD1($\times10^2$)↓ | CD2($\times10^5$)↓ | F1↑ | NC↑ | NR↓ |
|---|---|---|---|---|---|
| Dense Pairwise Prediction | 0.283 | 0.540 | 0.951 | 0.988 | 1.305 |
| Edge-Factorized | 0.283 | 0.540 | 0.951 | 0.988 | 1.311 |

**Edge-factorized Parameterization Effectiveness.** To validate the effectiveness of our edge-factorized parameterization, we compare it against a dense pairwise prediction baseline. The dense baseline directly regresses the $K \times K$ connectivity matrix without any low-rank constraints or edge probabilities. As shown in Table 5, the dense pairwise prediction baseline requires 6.08M parameters, whereas our edge-factorized design uses only 0.35M parameters, corresponding to 5.8% of the baseline. Despite this 94.2% reduction in parameter count, Table 6 shows that our design achieves comparable reconstruction quality on the ABC dataset. This indicates that dense connectivity regression is unnecessary and that enforcing a structured factorization provides an effective inductive bias that captures the essential geometry of local triangle formation, with substantially lower model complexity. The result supports our design choice to model connectivity using decomposable unary and pairwise terms rather than unconstrained dense prediction. We provide additional analyses in the appendix, including the comparison with OffsetOPT's triangle prediction network and robustness experiments under Gaussian noise and linear non-uniform sampling in Appendix D, as well as additional efficiency analysis in Appendix E.

## 5. Limitations

FastSESR still has several limitations. First, its performance depends on relatively reliable local neighborhood structures. To further analyze this issue, we provide additional robustness experiments with Gaussian noise and linear non-uniform sampling in Appendix F. As shown in Table 10, FastSESR remains stable under linear non-uniform sampling but degrades more noticeably under Gaussian noise. Table 11 further shows that FastSESR and OffsetOPT exhibit similar degradation trends under strong noise, suggesting that noise sensitivity is a common issue for explicit reconstruction methods based on local triangle candidate prediction.

Second, scene-level cross-dataset generalization remains challenging due to domain gaps in geometric layout, scale

distribution, and sampling patterns. Future work may improve robustness by incorporating denoising, noise-aware local priors, or stronger cross-dataset training strategies.

## 6. Conclusion

We presented FastSESR, a two-stage framework primarily for scene-level explicit surface reconstruction, replacing the costly test-time optimization paradigm with a single feed-forward inference, reducing surface reconstruction latency from minutes to seconds while preserving high geometric fidelity. We introduce a triangle candidate network to estimate triangle formation probabilities within local neighborhoods, enabling effective extraction of surface triangles, and an off-set optimization network that learns to predict offsets in an unsupervised manner using signals derived from the triangle candidate network. Together, these components provide an efficient and scalable pipeline for large-scale point clouds, offering a practical trade-off between speed and quality for scene-level explicit surface reconstruction.

## Acknowledgements

This work was supported by the Beijing Natural Science-Xiaomi Innovation Joint Foundation(No. L253007).

## Impact Statement

This paper presents work whose goal is to advance the field of machine learning. There are many potential societal consequences of our work, none of which we feel must be specifically highlighted here.

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

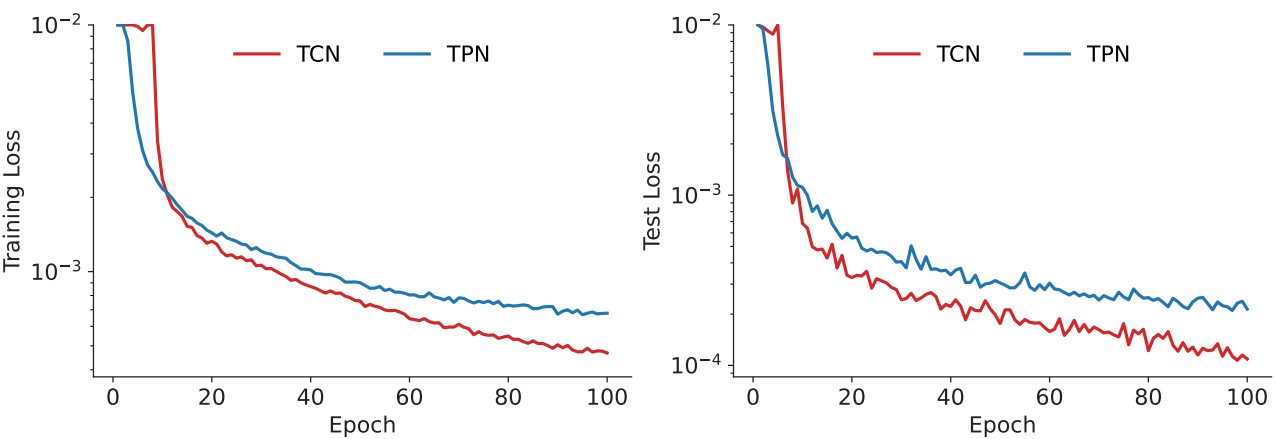

*Figure 4.* Comparison between our triangle candidate network (TCN) and triangle prediction network (TPN) of OffsetOPT (Lei, 2025). The red line refers to TCN, and the blue one refers to TPN of OffsetOPT.

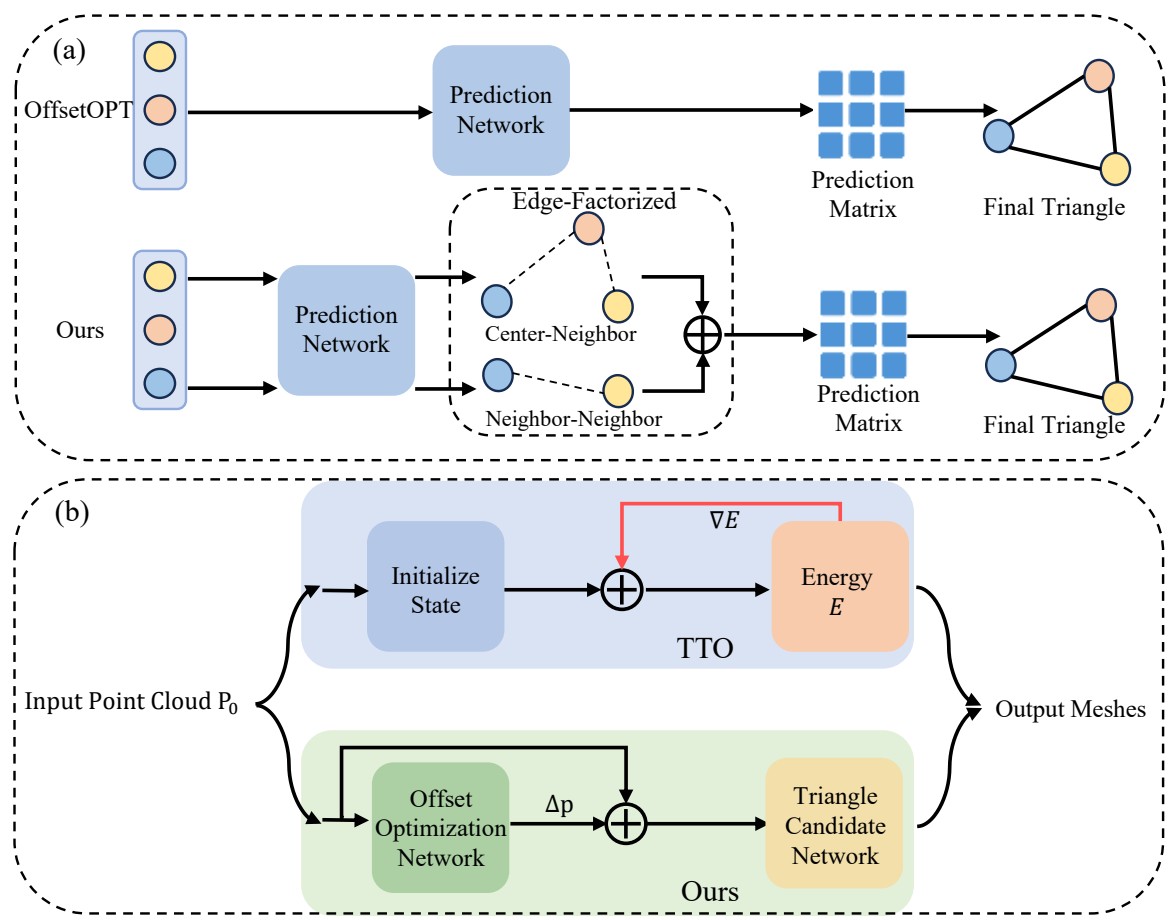

*Figure 5.* (a) Comparison between the triangular candidate network of FastSESR and the triangular prediction network of OffsetOPT. (b) Comparison between our learning-based surface reconstruction approach and test-time optimization methods, primarily represented by OffsetOPT (Lei, 2025) and Point2Mesh (Hanocka et al., 2020).

## A. Dataset Specifics and Partitioning

The ABC dataset (Koch et al., 2019) serves as the basis for our triangle candidate network, providing a collection of clean, synthetic meshes with high-quality, nearly equilateral triangulation. Following the protocols in CircNet (Lei et al., 2023) and OffsetOPT (Lei, 2025), we use 9,026 voxelized meshes, partitioned into 25% for training and 75% for testing. For the offset optimization network, we target diverse domains, including the test splits of ScanNet (Dai et al., 2017) and Matterport3D (Chang et al., 2017), as well as 10 random scenes from CARLA (Dosovitskiy et al., 2017); these data are provided by OffsetOPT (Lei, 2025). The object-level datasets are notably small, with FAUST (Bogo et al., 2014) and MGN (Bhatnagar et al., 2019) containing only 100 and 154 samples, respectively. To mitigate overfitting and ensure robust performance estimation despite limited data, we adopt a $K$-fold cross-validation strategy across all three large-scale scene datasets (ScanNet, Matterport3D, CARLA). Our data were obtained by calculating the mean value across three different fold division strategies.

## B. Evaluation Metrics

We evaluate the overall geometric quality of each reconstructed mesh using Chamfer Distance (CD1) and squared Chamfer Distance (CD2), and assess the quality of surface normals using Normal Consistency (NC). Following the common practice (Lei et al., 2023; Lei, 2025), we uniformly sample the same number of points (e.g., $10^5$) on the ground-truth mesh $\mathcal{T}$ and the reconstructed mesh $\tilde{\mathcal{T}}$. Let the resulting point clouds be $\mathcal{G}$ (ground truth) and $\mathcal{R}$ (reconstruction), respectively, and let each sampled point be associated with a unit normal.

**Chamfer Distance.** The Chamfer distance (CD1) is computed as

$$\text{CD1}(\mathcal{G}, \mathcal{R}) = \frac{1}{|\mathcal{G}|} \sum_{x \in \mathcal{G}} \min_{y \in \mathcal{R}} \|x - y\|_2 + \frac{1}{|\mathcal{R}|} \sum_{x \in \mathcal{R}} \min_{y \in \mathcal{G}} \|x - y\|_2. \tag{24}$$

The first term measures the completeness of the reconstruction (how well $\mathcal{R}$ covers $\mathcal{G}$), while the second term measures its accuracy (how close $\mathcal{R}$ is to $\mathcal{G}$). For squared Chamfer distance (CD2), $\|x - y\|_2$ is replaced by $\|x - y\|_2^2$.

**Normal Consistency and Normal Reconstruction Error in Degrees.** Let $n_x$ denote the unit normal at point $x$. We compute NC by averaging the absolute cosine similarity between normals of each point and its nearest neighbor on the other set:

$$\text{NC}(\mathcal{G}, \mathcal{R}) = \frac{1}{2} \left( \frac{1}{|\mathcal{G}|} \sum_{x \in \mathcal{G}} |n_x \cdot n_{y^*(x)}| + \frac{1}{|\mathcal{R}|} \sum_{x \in \mathcal{R}} |n_x \cdot n_{z^*(x)}| \right), \tag{25}$$

where $y^*(x) = \arg\min_{y \in \mathcal{R}} \|x - y\|_2$ and $z^*(x) = \arg\min_{z \in \mathcal{G}} \|x - z\|_2$. Higher NC indicates better agreement of local surface orientations. NR measures the average angular error (in degrees) between the normals of corresponding points.

$$\text{NR}(\mathcal{G}, \mathcal{R}) = \frac{1}{2} \left( \frac{1}{|\mathcal{G}|} \sum_{x \in \mathcal{G}} \frac{180}{\pi} \arccos\big(|n_x \cdot n_{y^*(x)}|\big) + \frac{1}{|\mathcal{R}|} \sum_{x \in \mathcal{R}} \frac{180}{\pi} \arccos\big(|n_x \cdot n_{z^*(x)}|\big) \right). \tag{26}$$

**F-score.** For a distance threshold $\epsilon$, we compute F1 as the harmonic mean of recall and precision:

$$\text{F1}(\epsilon) = \frac{2 \, \text{Recall}(\epsilon) \, \text{Precision}(\epsilon)}{\text{Recall}(\epsilon) + \text{Precision}(\epsilon)}. \tag{27}$$

The recall and precision are calculated as

$$\text{Recall}(\epsilon) = \frac{1}{|\mathcal{G}|} \sum_{x \in \mathcal{G}} \mathbb{I}\left( \min_{y \in \mathcal{R}} \|x - y\|_2 < \epsilon \right), \quad \text{Precision}(\epsilon) = \frac{1}{|\mathcal{R}|} \sum_{x \in \mathcal{R}} \mathbb{I}\left( \min_{y \in \mathcal{G}} \|x - y\|_2 < \epsilon \right), \tag{28}$$

where $\epsilon$ is a small distance threshold and $\mathbb{I}(\cdot)$ is the indicator function. Recall measures the fraction of ground-truth points covered by the reconstruction within $\epsilon$, while precision measures the fraction of reconstructed points that lie within $\epsilon$ of the ground truth; higher F1 indicates a better balance between completeness and accuracy.

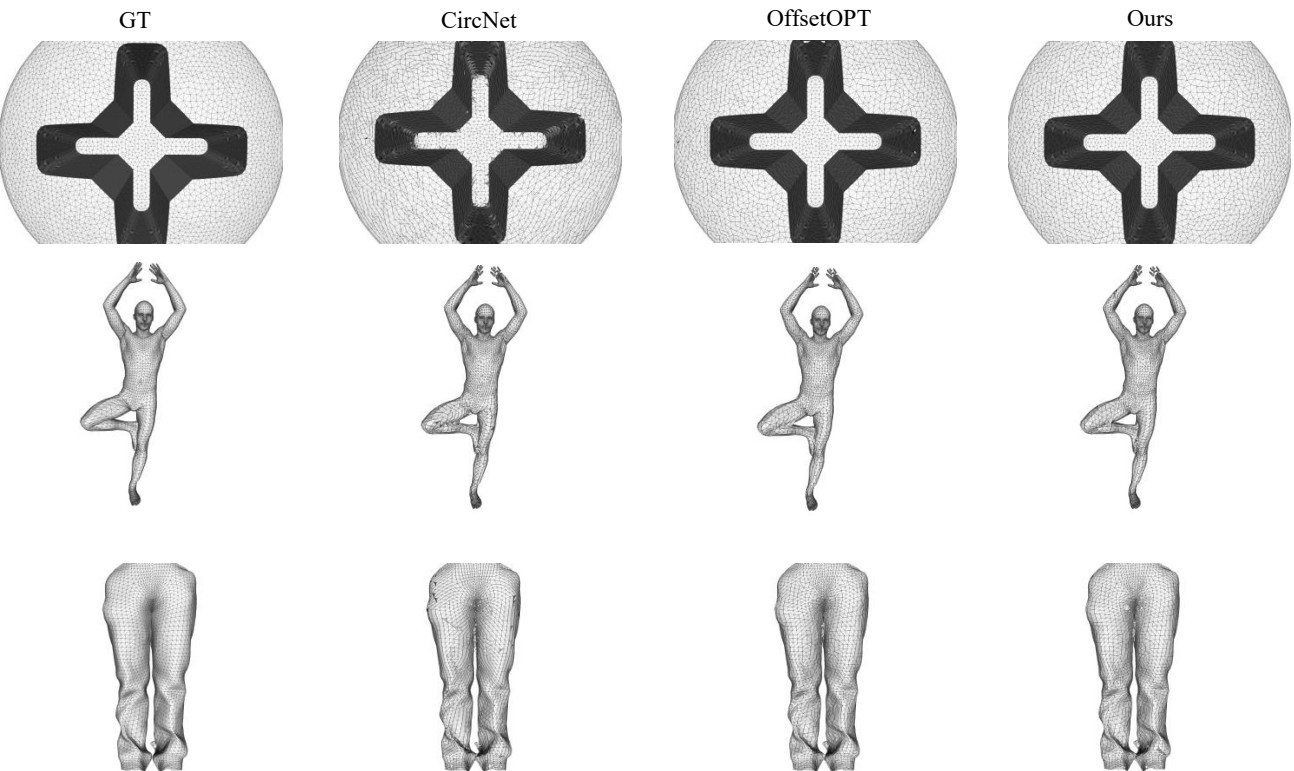

*Figure 6.* Visualization comparison between different methods. Each row represents: ABC, FAUST, MGN, respectively.

## C. More visualization

We provide additional qualitative results. Figure 6 shows the visual results on the shape-level dataset. Each row represents: ABC (Koch et al., 2019), FAUST (Bogo et al., 2014), MGN (Bhatnagar et al., 2019), respectively. It is evident that, although CircNet (Lei et al., 2023) exhibits slightly inferior performance characterized by the presence of more holes and geometric artifacts, all methods successfully reconstructed the point cloud data at the shape level with good surface reconstruction accuracy.

## D. Additional Ablation

**Triangle Candidate Network vs OffsetOPT's Triangle Prediction Network.** We compare our triangle candidate network (TCN) with OffsetOPT's triangle prediction network (TPN) (Lei, 2025), as shown in Figure 4 and Figure 5 (a). Our network achieves better results in both training and testing. In Figure 5 (b), we also compare our second-stage learning-based approach against test-time optimization methods, represented by OffsetOPT (Lei, 2025) and Point2Mesh (Hanocka et al., 2020).

*Table 7.* Ablation study on weight sharing strategy. We compare our recurrent weight-sharing design against an unrolled variant that uses independent parameters at each update step on CARLA (Dosovitskiy et al., 2017).

| Method | CD1($\times 10^2$)↓ | CD2($\times 10^5$)↓ | F1↑ | NC↑ | NR↓ |
|---|---|---|---|---|---|
| unshared | 0.115 | 0.178 | 0.990 | 0.967 | 4.965 |
| shared | **0.114** | **0.171** | **0.990** | **0.967** | **4.929** |

**Shared vs Unshared.** We compare our recurrent weight-sharing design against an unrolled variant that uses independent parameters at each update step on CARLA (Dosovitskiy et al., 2017). Weight sharing improves surface quality, indicating

*Table 8.* Comparison of model size and computation across different surface reconstruction methods with the same input point cloud.

| Method | Params(M) | MACs |
|---|---|---|
| CircNet | 6.76 | 41.79 G |
| PointTriNet | 1.51 | 86.47 G |
| OffsetOPT (S1) | 7.94 | 41.99 G |
| OffsetOPT (S2) | – | 4.20 T |
| OffsetOPT (Total) | 7.94 | 4.24 T |
| FastSESR (S1) | 0.34 | 424.34 G |
| FastSESR (S2) | 0.18 | 6.95 G |
| FastSESR (Total) | 0.53 | 431.29 G |

*Table 9.* Ablation study of upsampling strategies on Matterport3D.

| Method | CD1 $(\times 10^2) \downarrow$ | CD2 $(\times 10^5) \downarrow$ | F1 $\uparrow$ | NC $\uparrow$ | NR $\downarrow$ |
|---|---|---|---|---|---|
| Learnable upsampling (Ours) | **0.159** | **0.171** | **0.997** | **0.940** | 10.543 |
| Grad-PU (He et al., 2023) | 0.160 | 0.173 | 0.996 | 0.940 | **10.502** |

that the recurrent design provides a beneficial inductive bias for learning a stable geometric update operator, rather than relying on step-specific parameters.

**Upsampling Strategy.** To analyze the effect of the upsampling strategy in OON, we replace the learnable upsampling module with Grad-PU (He et al., 2023). Grad-PU is an arbitrary-scale point cloud upsampling method based on learned distance functions, and is used here as an alternative strategy for coarse-level offset propagation. As shown in Table 9, the two strategies achieve comparable reconstruction quality. Compared with Grad-PU, our learnable upsampling strategy performs slightly better on major metrics such as CD1, CD2, and F1, suggesting that it can more effectively integrate the multi-scale features and skip-connected features in OON for coarse-to-fine offset propagation.

# E. Efficiency Analysis

Table 8 compares model size (Params) and overall computation (Total MACs) across different explicit surface reconstruction methods under the same input point cloud. CircNet (Lei et al., 2023) and PointTriNet (Sharp & Ovsjanikov, 2020) contain 6.76M and 1.51M parameters, with total MACs of 41.79G and 86.47G, respectively; despite being lighter, PointTriNet incurs higher computation, consistent with iterative inference. For OffsetOPT (Lei, 2025), the first stage costs 41.99G MACs, whereas the second stage dramatically increases to 4.20T MACs due to iterative test-time optimization, resulting in 4.24T total MACs. This highlights that test-time optimization introduces a substantial inference-time burden, with the overall cost dominated by the second stage. In contrast, FastSESR adopts a lightweight two-stage design with only 0.53M parameters in total and reduces the total computation to 431.29G MACs. Its compute is mainly attributed to the first stage (424.34G), while the second stage contributes only a minor overhead (6.95G). The computational cost is primarily driven by two components within the TransGNN blocks: (i) message passing over $K \times K_{msg}$ edges, and (ii) quadratic self-attention within local patches. Despite this, FastSESR still maintains a significantly lower memory footprint compared to OffsetOPT.

# F. Additional Robustness Experiments

To further evaluate the stability of FastSESR under input perturbations, we conduct additional robustness experiments with Gaussian noise and linear non-uniform sampling. Gaussian noise is added to the normalized input point cloud as:

$$\tilde{\mathbf{x}}_i = \mathbf{x}_i + \boldsymbol{\epsilon}_i, \qquad \boldsymbol{\epsilon}_i \sim \mathcal{N}(\mathbf{0}, \sigma^2 \mathbf{I}_3), \tag{29}$$

where $\mathbf{x}_i$ denotes a normalized input point, $\tilde{\mathbf{x}}_i$ is the perturbed point, and $\sigma$ controls the noise level.

For linear non-uniform sampling, we assign a coordinate-dependent retention probability along a spatial direction. Given

*Table 10.* Robustness evaluation on CARLA under Gaussian noise and linear non-uniform sampling.

| Setting | CD1 $(\times 10^2)\downarrow$ | CD2 $(\times 10^5)\downarrow$ | F1 $\uparrow$ | NC $\uparrow$ | NR $\downarrow$ | HD $\downarrow$ |
|---|---|---|---|---|---|---|
| FastSESR (clean) | **0.114** | **0.171** | **0.990** | **0.967** | **4.929** | 0.044 |
| Gaussian noise, $\sigma = 0.005$ | 0.522 | 2.274 | 0.618 | 0.572 | 51.456 | 0.028 |
| Gaussian noise, $\sigma = 0.01$ | 0.881 | 7.559 | 0.395 | 0.533 | 54.646 | **0.042** |
| Linear non-uniform sampling | 0.115 | 0.182 | 0.990 | 0.966 | 5.103 | 0.043 |

*Table 11.* Robustness comparison under varying Gaussian noise levels on the ABC dataset.

| Method | Noise Level | CD1 $(\times 10^2)\downarrow$ | CD2 $(\times 10^5)\downarrow$ | F1 $\uparrow$ | NC $\uparrow$ | NR $\downarrow$ |
|---|---|---|---|---|---|---|
| CircNet | $\sigma = 0$ | 0.284 | 0.544 | 0.950 | 0.985 | 1.758 |
| OffsetOPT | $\sigma = 0$ | 0.283 | 0.540 | 0.951 | 0.988 | 1.318 |
| FastSESR (Ours) | $\sigma = 0$ | 0.283 | 0.540 | 0.951 | 0.988 | 1.311 |
| CircNet | $\sigma = 0.1$ | 0.328 | 0.720 | 0.908 | 0.965 | 10.053 |
| OffsetOPT | $\sigma = 0.1$ | 5.880 | 332.094 | 0.058 | 0.522 | 55.574 |
| FastSESR (Ours) | $\sigma = 0.1$ | 5.888 | 338.663 | 0.060 | 0.518 | 55.765 |
| CircNet | $\sigma = 0.2$ | 0.419 | 1.245 | 0.793 | 0.931 | 16.735 |
| OffsetOPT | $\sigma = 0.2$ | 8.735 | 724.544 | 0.042 | 0.513 | 56.272 |
| FastSESR (Ours) | $\sigma = 0.2$ | 8.727 | 733.308 | 0.043 | 0.511 | 56.339 |

the coordinate $a_i$ of point $\mathbf{x}_i$ along this direction, we first normalize it as:

$$t_i = \frac{a_i - a_{\min}}{a_{\max} - a_{\min} + \epsilon}. \tag{30}$$

The retention probability is then defined by a linear function:

$$p_i = p_{\min} + (p_{\max} - p_{\min})t_i. \tag{31}$$

Finally, the non-uniform point cloud is obtained through Bernoulli sampling:

$$m_i \sim \text{Bernoulli}(p_i), \qquad \mathbf{x}_i \text{ is kept if } m_i = 1. \tag{32}$$

The results on CARLA are reported in Table 10. We further compare FastSESR with CircNet and OffsetOPT under different Gaussian noise levels on the ABC dataset, shown in Table 11.

The results show that FastSESR remains stable under linear non-uniform sampling, with most metrics close to the clean-input setting. In contrast, Gaussian noise causes more noticeable degradation by corrupting local geometric structures. The comparison on ABC further shows that FastSESR and OffsetOPT exhibit similar degradation trends under strong noise, while CircNet is more stable in noisy settings. These results suggest that FastSESR is robust to moderate density variation but remains sensitive to strong random noise.

