# OpenReview forum: "FastSESR: Fast Scene-level Explicit Surface Reconstruction"
_ICML.cc/2026/Conference — ICML 2026 regular_

### Official Review · Reviewer_Z1yY · 2026-03-04

**Soundness:** 3
**Presentation:** 3
**Significance:** 3
**Originality:** 3
**Overall Recommendation:** 4
**Confidence:** 5

**Summary:**

This paper proposes a two-stage explicit surface reconstruction framework. The method combines triangle candidate estimation with an offset optimization network to accelerate mesh reconstruction from point clouds. Experimental results demonstrate that the proposed approach achieves competitive reconstruction quality while significantly reducing computational cost.

**Compliance With Llm Reviewing Policy:**

Affirmed.

**Final Justification:**

All my concerns have been well addressed, and I will maintain my positive score.

**Key Questions For Authors:**

See Weaknesses.

**Limitations:**

Yes.

**Strengths And Weaknesses:**

### Strengths

1. The paper proposes a two-stage explicit surface reconstruction framework.

2. It combines triangle candidate estimation with an offset-based optimization network, which substantially reduces the number of optimization iterations and model parameters.

3. Experimental results show that the method achieves performance comparable to previous approaches while providing clear computational advantages.

### Weaknesses

1. In Equation (1), the denominator uses the nearest point \(p_1\) for normalization instead of the farthest point \(p_K\). When \(p_1\) is extremely close to the center point, this normalization may lead to scale instability.

2. The descriptions around Equations (6)–(8) are somewhat difficult to follow and partially repetitive. It is unclear what features are used to compute the scores. In particular, \(Y\) and \(X_j\) are not explicitly defined in the text.

3. In Equations (7), (9), and (12), the indexing conventions for points (\(i,j,k\)) and network layers (\(t,l\)) are somewhat inconsistent. It would improve readability if these notations were unified and clearly distinguished.

4. The method uses interpolation to recover the point cloud resolution [1]. However, interpolation-based propagation of offsets may smooth the deformation field and potentially blur fine geometric details. It would be helpful if the authors discussed the potential limitations of this design.

### References

[1] Grad-PU: Arbitrary-Scale Point Cloud Upsampling via Gradient Descent With Learned Distance Functions.

---

> ### Author Rebuttal · Authors · 2026-03-30
>
> Thanks for the careful reading and constructive suggestions. We address each point below and will revise the paper accordingly.
>
> **(1) Normalization in Eq. (1).**
> We agree that normalizing by the nearest neighbor can be potentially unstable when \(p_1\) is extremely close to the center point. In our setting, however, the stage-1 triangle network (TCN) is trained on the ABC dataset, whose point clouds are dense and approximately uniformly sampled, so such degenerate cases are rare in practice. For this reason, both nearest-point and farthest-point normalization are workable under our training distribution. To clarify this issue, we additionally tested normalization by the farthest neighbor \(p_K\). The new results show that nearest-point normalization is slightly better overall.  We provide a visualization of comparing these two methods in the second picture in https://anonymous.4open.science/r/rebuttal_tab_pict-41B8/. We will report this comparison in the revision.
>
> **(2) Clarity of Eqs. (6)–(8).**
> The current presentation is indeed unclear, mainly because the feature definitions were omitted. Here, \(X_j\) denotes the feature of neighbor \(j\) after the TransGNN block, and \(Y\) denotes the TransGNN output features of the local patch. The unary score \(s_j\) is predicted from \(Y\) by a lightweight MLP, while the pairwise term \(s_{jk}\) is computed from projected neighbor features. The final score is formed as \(S_{jk}=s_j+s_k+s_{jk}\). We will revise this part to define all variables explicitly and rewrite Eqs. (6)–(8) more clearly.
>
> **(3) Inconsistent indexing notation.**
> We agree that the current index convention is not fully consistent. In the final version, we will unify the notation for point indices \((i,j,k)\) and network layer indices \((t,l)\), and explicitly distinguish spatial indices from iteration or hierarchy indices throughout the method section. We expect this change to substantially improve readability.
>
> **(4) Interpolation-based offset propagation may oversmooth details.**
> Our coarse-to-fine decoder uses interpolation to restore point resolution, and such offset propagation may indeed smooth the deformation field and potentially weaken very fine geometric details. We will add this discussion to the limitation section.  We also conducted an additional experiment by replacing our interpolation module with Grad-PU[1]. On Matterport3D, the resulting average metrics are:
>
> | Method | CD1($\times 10^{2}$)↓ | CD2($\times 10^{5}$)↓ | F1↑ | NC↑ | NR↓ | HD↓ |
> |---|---:|---:|---:|---:|---:|---:|
> | learnable interpolation (Ours) | 0.159 | 0.171 | 0.997 | 0.940 | 10.543 | 0.011 |
> | Grad-PU  | 0.160 | 0.173 | 0.996 | 0.940 | 10.502 |0.012 |
>
> These results suggest that Grad-PU and our learnable interpolation achieve broadly similar reconstruction quality within our framework. Interestingly, we found that Grad-PU converges faster during training, which can reduce offline training cost. We will add this comparison and discuss the trade-off more explicitly in the revision, and add references to Grad-PU into the paper.
>
> References
>
> [1] Grad-PU: Arbitrary-Scale Point Cloud Upsampling via Gradient Descent With Learned Distance Functions

---

> > ### Author Rebuttal · Reviewer_Z1yY · 2026-04-03
> >
> > Thanks for the author's response. All my concerns are well addressed.

---

### Official Review · Reviewer_6w2m · 2026-03-10

**Soundness:** 4
**Presentation:** 3
**Significance:** 2
**Originality:** 3
**Overall Recommendation:** 4
**Confidence:** 4

**Summary:**

This paper targets at triangular mesh reconstruction from dense 3D point cloud input. A two-stage framework called FastSESR is designed for reconstructing scene-level surfaces. Compared to the traditional time-consuming test-time optimization paradigm, the proposed method is much more efficient while maintaining competitive reconstruction quality. The core idea involves a coarse-to-fine offset-driven refinement workflow. Experiments show the good performance of the proposed method.

**Compliance With Llm Reviewing Policy:**

Affirmed.

**Final Justification:**

The authors' responses have addressed most of my concerns.

**Key Questions For Authors:**

It seems that the proposed method can be applied to object-level data without much difficulty. Why do the authors particularly restrict the scope of this paper to the scene level?

**Limitations:**

1. lack of technical novelty
2. overly idealized assumption about the input data
3. insufficient experimental comparison

**Strengths And Weaknesses:**

### Strengths:
1. The overall processing pipeline achieves significant speedup without substantial quality degradation.
2. The two-stage coarse-to-fine strategy is technically sound.
3. The proposed edge-factorized parameterization is a smart solution for efficiency.

### Weaknesses:
1. It is unclear how many points are contained in the input point cloud and how the input point cloud is obtained.
2. The proposed method is only applied on perfect testing cases, i.e., noise-free, dense, uniform. This setting makes the actual task much less challenging. Moreover, the provided experimental results did not reflect the ability of FastSESR to capture high-frequency geometric details.
3. There is a notable absence of comparisons with state-of-the-art implicit surface reconstruction approaches, which have been extensively studied over the recent few years and show highly competitive performance.

---

> ### Author Rebuttal · Authors · 2026-03-30
>
> We thank the reviewer for the careful reading and constructive feedback. We clarify the main points below.
>
> ## (1) Input point count and data acquisition.
>
> We agree that the paper should state the input setting more explicitly. For shape-level reconstruction, the default input point cloud is the set of vertices of the test mesh, following OffsetOPT and CircNet[1]. We report the maximum number of inputs below:
> | Dataset |  Maximum Number of Input Points |
> |---|-----:|
> | ABC | 19,669 |
> | FAUST | 6,890 |
> | MGN | 10,116 |
>
> For scene-level reconstruction, following OffsetOPT, we use 1M input points per scene: for ScanNet and Matterport3D, these points are randomly sampled from the ground-truth meshes; for CARLA, they are randomly sampled from the provided dense point clouds. Before reconstruction, we voxelize the input to regularize sampling density, using a grid size of 2 cm for ScanNet and 10 cm for Matterport3D/CARLA. We will add these details clearly to the experimental setup.
>
> ## (2) Idealized input assumptions, robustness, and high-frequency details.
>
> We agree that the current paper mainly emphasizes relatively clean dense inputs, because our goal is to study whether explicit surface reconstruction can be scaled effectively to scene-level data while preserving quality and achieving practical speed. We add experiments under both different noise levels and non-uniform input sampling. Specifically, we test the method by perturbing normalized inputs with i.i.d. isotropic Gaussian noise, following CircNet[1] and PINC[2]：
>  $$
> x_i' = x_i + \epsilon_i,\qquad \epsilon_i \sim \mathcal{N}(0,\sigma^2 I),
> $$
> and by constructing non-uniformly sampled point clouds.
>  Results on CARLA are below:
> | Setting               |  CD1($\times 10^{2}$)↓ |  CD2($\times 10^{5}$)↓ |   F1↑ |   NC↑ |   NR↓ |   HD↓ |
> | --------------------- | ----: | ----: | ----: | ----: | ----: | ----: |
> | FastSESR (Ours)   | 0.114 | 0.171 | 0.990 | 0.967 | 4.929 |     0.044|
> | Gaussian noise, σ=0.005 | 0.522 | 2.274 | 0.618 | 0.572 | 51.456 |0.028 |
> | Gaussian noise, σ=0.01 |   0.881 |   7.559 |   0.395 |   0.533 |   54.646 |   0.042 |
> | Non-uniform sampling  | 0.115 | 0.182 | 0.990 | 0.966 | 5.103 | 0.043 |
>
> These results suggest that FastSESR remains robust to non-uniform sampling, with only marginal degradation compared with the original input. In contrast, the performance drops noticeably under Gaussian noise, especially as the noise level increases, indicating that noise remains a clear limitation of the current method.
>
> Regarding high-frequency geometric details, our goal is to improve scene-level reconstruction speed while maintaining competitive quality, rather than to claim universally superior detail recovery. We therefore additionally report Edge-CD1 and Edge-F1, following CircNet and OffsetOPT. These metrics focus on points sampled near edges and corners. On CARLA, FastSESR outperforms OffsetOPT and NKSR on these detail-oriented metrics, suggesting that the large speedup does not come at the cost of degraded fine geometric details. Here ‘+n’ indicates the usage of ground-truth normals.
>
> | Method | ECD1($\times 10^{2}$)↓ | EF1↑ |
> |---|---:|---:|
> | NKSR (+n) | 0.679 | 0.903 |
> | NKSR | 1.013 | 0.810 |
> | OffsetOPT | 0.317 | 0.946 |
> | FastSESR (Ours) | 0.290 | 0.953 |
>
> ## (3) Comparison with implicit surface reconstruction methods.
>
> We respectfully note that the paper already includes comparisons with representative implicit methods, namely SPSR and NKSR, on both scene-level and shape-level benchmarks. These baselines are included because they are strong and widely used references for large-scale reconstruction, even though some variants require normals. Here, we add a comparison of the ABC dataset with the latest implicit surface reconstruction methods.
> | Method | CD1($\times 10^{2}$)↓ | NC↑ | F1↑ |
> |---|---:|---:|---:|
> | SALS [3] (ICLR 2025) | 0.456 | 0.973 | 0.630 |
> | FastSESR (ours)| 0.283 | 0.988 | 0.951 |
>
> ## (4) Motivation for focusing on scene-level reconstruction
>
>  We focus on scene-level reconstruction because this is where the main bottleneck of prior explicit methods becomes most severe: optimization-based approaches such as OffsetOPT achieve strong quality, but their per-scene test-time optimization is prohibitively slow on large-scale data. At the same time, we do not claim the method is restricted to scenes: we also evaluate on ABC, FAUST, and MGN, where it shows good generalization and competitive performance at the shape level as well. In other words, the contribution is not to exclude object-level reconstruction, but to address the more challenging and less explored scene-level explicit reconstruction setting while retaining good shape-level performance.
>
> reference：
>
> [1]Circnet: Meshing 3d point clouds with circumcenter detection.ICLR 2023.
>
> [2]PINC: p-Poisson surface reconstruction in curl-free flow from point clouds.NeurIPS 2023.
>
> [3]Shape as line segments: Accurate and flexible implicit surface representation.ICLR 2025.

---

### Official Review · Reviewer_AT5g · 2026-03-12

**Soundness:** 3
**Presentation:** 3
**Significance:** 3
**Originality:** 3
**Overall Recommendation:** 5
**Confidence:** 4

**Summary:**

This paper presents a novel method for explicit safe reconstruction that is considerably faster than the recently proposed OffsetOPT. The proposed approach achieves comparable results on the ABC dataset, slightly better performance on CARLA, and nearly on par (though slightly worse) performance on the remaining datasets.
The method combines a Triangular Candidate Network (TCN), which predicts the existence of local triangles, and an Offset Optimization Network, which refines the positions of points. This refinement stage enables the method to generalize beyond the initial training performed on the ABC dataset. The evaluation compares the proposed method with existing approaches across multiple datasets, using several metrics, and reports both reconstruction quality and runtime performance.

**Compliance With Llm Reviewing Policy:**

Affirmed.

**Final Justification:**

The authors fully addressed all my concerns in the rebuttal response with clarifications and new experiments. I urge the authors to include these new details and experiments in the final version of the paper to make it clear for the reader.

Particularly the clarifications on how execution times are measured and the additional experiments on noise significantly improve the quality of the work.

I have raised my score.

**Key Questions For Authors:**

1) Execution times:
1.1) What are the full machine specifications used for the experiments (CPU, memory, operating system)? The GPU is already specified in the paper.
1.2) Were both OffsetOPT and FastSESR executed on the same machine for the results reported in Table 3? Do both methods utilize the GPU?
1.3) What frameworks or libraries were used to implement OffsetOPT and FastSESR (e.g., PyTorch, CUDA, custom C++), and were comparable optimizations applied? Were the same computational optimizations applied to both implementations? It might be the case that the authors are comparing a non-optimized version of OffsetOPT with an optimized version of FastSESR, and that the difference in execution time is derived mostly from that.
1.4) How does the proposed method compare in execution time with other competitive approaches, such as CircNet, PointTriNet, and DSE?
1.5) How does the method scale with point cloud size in terms of memory usage and execution time? How does this scaling compare with existing approaches?
1.6) In Appendix E, what does the acronym MACs refer to? Does it denote Multiply-Accumulate operations? It is interesting that methods such as CircNet and PointTriNet have a much lower number of MACs, although MACs is likely not a good indicator of performance.

2) Quantitative Evaluation:
2.1) Although the paper explicitly mentions noise handling as a limitation, how much noise can the proposed method tolerate? It would be helpful to include an experiment that varies the noise level and evaluates the resulting reconstruction quality using representative metrics.
2.2) Could additional metrics be included in Tables 1 and 2, such as Hausdorff Distance and mesh-quality metrics for shape-level datasets, as well as accuracy and completeness metrics for scene-level datasets?
2.3) How sensitive is the method to training on the ABC dataset? Have the authors evaluated generalization when training on different datasets or subsets?

3) Minor Comments:
* The notation should be used consistently throughout the paper. For instance, matrices are typically represented using bold capital letters and vectors using bold lowercase letters, but this convention is not always followed (e.g., Equation 8).
* Table 5 is difficult to read due to the small font size. Please increase the font size; using two columns for the table header or method names may help if space is limited.

**Limitations:**

Yes

**Strengths And Weaknesses:**

Strengths:
+ The paper is clearly written and easy to follow, with appropriate references to related work.
+ The proposed architecture combines triangle prediction with offset optimization in a novel pipeline.
+ The method achieves substantially faster inference compared with OffsetOPT while maintaining comparable reconstruction quality.
Weaknesses:
- The reconstruction results are, at best, comparable to the current state of the art. Consequently, the main contribution of the work appears to lie primarily in improved computational performance.
- Since computational efficiency is the primary claimed advantage, the experimental methodology used to measure execution time is not sufficiently detailed to ensure a fair comparison.
- Execution time comparisons are reported only against OffsetOPT, while other relevant competing approaches are not included.
- The quantitative evaluation could be strengthened by including additional metrics, such as Hausdorff Distance and mesh-quality metrics for shape-level datasets, as well as accuracy and completeness metrics for scene-level datasets.
- Although the authors explicitly mention noise handling as a limitation of the approach, it would be helpful to provide a quantitative evaluation of the level of noise that the method can currently tolerate.
Overall, while the efficiency improvements are promising, the experimental evaluation would benefit from additional comparisons and more detailed reporting of runtime conditions.

---

> ### Author Rebuttal · Authors · 2026-03-30
>
> Thanks for the detailed comments. Our main claim is comparable reconstruction quality with substantially improved efficiency, rather than superior accuracy on every metric.
>
> ## 1. Runtime protocol and fairness of efficiency comparison
>
> (1.1~1.3) Machine configuration
> All timing results were obtained on the same machine: dual Intel Xeon Gold 5218R @ 2.10GHz CPU, 251 GiB RAM, Ubuntu 20.04, NVIDIA RTX 3090 24GB, PyTorch 2.6.0 / CUDA 11.8. OffsetOPT and FastSESR were run on the same machine under the same hardware conditions, and both used GPU. We used OffsetOPT’s official code without modification. Neither method used custom CUDA/C++ kernels or special engineering optimizations. Hence, the measured speedup is not due to comparing an unoptimized baseline against an optimized version of our method.
>
> (1.4) We mainly compare the runtime with OffsetOPT because it is the most relevant prior explicit scene-level reconstruction method and the closest baseline in formulation. By contrast, CircNet, PointTriNet, and DSE are primarily evaluated on shape-level reconstruction and do not report comparable scene-level runtime. For completeness, we additionally include the PointTriNet runtime below and will clarify this more explicitly in the revision.
> | Dataset | PointTriNet (s) | OffsetOPT (s) | FastSESR (s) |
> |---|---:|---:|---:|
> | ScanNet | — | 599.47 | 24.43 |
> | Matterport3D | 320.12 | 854.31 | 39.04 |
> | CARLA | 305.03 | 842.16 | 32.36 |
>
>
> (1.5) Scalability with Point Cloud Size on CARLA
>
> Reconstruction Time(s)
>
> | Number of Points | FastSESR | PointTriNet|
> |---:|---:|---:|
> | 100,000 | 9.28 | 14.46 |
> | 150,000 | 15.16 | 22.89 |
> | 200,000 | 21.84 | 31.11 |
>
> Peak Memory(MB)
>
> | Number of Points | FastSESR | PointTriNet |
> |---:|---:|---:|
> | 100,000 | 4674.31 | 2466.71 |
> | 150,000 | 4773.89 | 2526.99 |
> | 200,000 | 4875.99 | 2568.59 |
>
> Both runtime and memory increase with point cloud size.
>
> (1.6) MACs refer to multiply-accumulate operations. We will define this explicitly in the appendix. We also agree that MACs are only a rough proxy and do not fully predict wall-clock speed.
>
>
> ## 2. Quantitative evaluation
>
> (2.1) Noise tolerance
>
> We additionally conducted experiments under different noise levels and non-uniform sampling. Specifically, we test the method by perturbing normalized inputs with i.i.d. isotropic Gaussian noise, following CircNet (ICLR2023) and PINC (NeurIPS 2023)：
>  $$
> x_i' = x_i + \epsilon_i,\qquad \epsilon_i \sim \mathcal{N}(0,\sigma^2 I),
> $$
> and by constructing non-uniformly sampled point clouds.  Results on CARLA are below:
> | Setting               |  CD1($\times 10^{2}$)↓ |  CD2($\times 10^{5}$)↓ |   F1↑ |   NC↑ |   NR↓ |   HD↓ |
> | --------------------- | ----: | ----: | ----: | ----: | ----: | ----: |
> | FastSESR (Ours)   | 0.114 | 0.171 | 0.990 | 0.967 | 4.929 |     0.044|
> | Gaussian noise, σ=0.005 | 0.522 | 2.274 | 0.618 | 0.572 | 51.456 |0.028 |
> | Gaussian noise, σ=0.01 |   0.881 |   7.559 |   0.395 |   0.533 |   54.646 |   0.042 |
> | Non-uniform sampling  | 0.115 | 0.182 | 0.990 | 0.966 | 5.103 | 0.043 |
>
> (2.2) Additional metrics
>
> We have now added HD in the supplementary robustness experiments. For the main evaluation, our current metrics follow common practice in surface reconstruction papers such as CircNet, NKSR, and DSE, and already cover geometric fidelity and reconstruction quality from multiple aspects. In particular, for shape-level datasets, CD1/CD2 and F1 evaluate geometric fidelity, while NC and NR reflect surface quality. For scene-level datasets, the two directional terms in CD1/CD2 correspond to completeness and accuracy, respectively, and F1 further summarizes the precision/recall trade-off.
>
> (2.3) Sensitivity to ABC training data and generalization
>
> Our method already shows reasonable shape-level generalization: OON is trained on ABC and directly transferred to FAUST and MGN, as discussed in Sec. 4.2. You can also see the table in the first picture of https://anonymous.4open.science/r/rebuttal_tab_pict-41B8/. We also observed that cross-dataset generalization at the scene level is more challenging, likely because scene datasets exhibit much larger domain gaps in geometry and layout. For example, training on CARLA and testing on Matterport3D gives:
> | Setting               |  CD1($\times 10^{2}$)↓ |  CD2($\times 10^{5}$)↓ |   F1↑ |   NC↑ |   NR↓ |   HD↓ |
> | --------------------- | ----: | ----: | ----: | ----: | ----: | ----: |
> | CARLA->CARLA   | 0.114 | 0.171 | 0.990 | 0.967 | 4.929 |     0.044 |
> | CARLA->Matterport3D | 0.159 | 0.172 | 0.997 | 0.940 | 10.555 |0.011 |
>
> ## 3. Minor comments
>
> We will revise the paper to use more consistent notation throughout, including Eq. 8. We will enlarge the font in Table 5 and reformat it for readability.
>
> ## Reference
> CircNet: Meshing 3D Point Clouds with Circumcenter Detection. ICLR 2023.
>
> PINC: p-Poisson Surface Reconstruction in Curl-Free Flow from Point Clouds. NeurIPS 2023.

---

> > ### Author Rebuttal · Reviewer_AT5g · 2026-04-02
> >
> > Thank you for the response. I appreciate the clarification regarding the runtime conditions. The additional runtime and scalability analysis with PointTriNet is also helpful for better contextualizing efficiency.
> >
> > That said, since efficiency is a central contribution of the paper, I still believe it would strengthen the work to better position FastSESR with respect to other fast reconstruction approaches beyond scene-level methods. In particular, CircNet is an efficient mesh reconstruction and reports strong runtime performance alongside competitive quality. While I understand that it is primarily evaluated at the shape level, it addresses a closely related reconstruction problem. A qualitative discussion or approximate comparison of runtime and reconstruction characteristics would clarify how FastSESR fits within the state of art.
> >
> > Regarding robustness, I appreciate the additional noise experiments. However, since the paper explicitly mentions sensitivity to noise as a limitation, it would be helpful to better contextualize the magnitude of this degradation (and whether it may be a problem in some applications). In particular, could the authors provide a comparative evaluation against methods CircNet and OffsetOPT under similar noise settings? This would make it clearer whether the observed performance drop is significant relative to prior work, or within the expected range for this class of methods.

---

> > > ### Author Response · Authors · 2026-04-03
> > >
> > > Thank you for your constructive feedback. We appreciate the opportunity to better position FastSESR against CircNet and to provide a clearer context regarding our method's robustness to noise.
> > >
> > > ## Comparison with CircNet (Efficiency and Reconstruction Characteristics)
> > >
> > > To clarify how FastSESR fits within the state-of-the-art regarding efficiency, we provide a direct runtime comparison on the ABC dataset and Matterport3D dataset under the same conditions.
> > >
> > > Table 1: Runtime Comparison
> > >
> > > | Dataset | Method | Average Runtime per Shape/Scene (s) |
> > > | --- | --- | ---: |
> > > | ABC | CircNet | 3.47 |
> > > | ABC | FastSESR (Ours) | 1.27 |
> > > | Matterport3D | CircNet | 78.45 |
> > > | Matterport3D | FastSESR (Ours) | 39.04 |
> > >
> > > As shown in Table 1, FastSESR is about 2.7× faster than CircNet on ABC and about 2.0× faster on Matterport3D.
> > >
> > > In terms of reconstruction characteristics, the two methods are optimized for different challenges. CircNet relies on circumcenter detection, which makes it highly effective for shape-level meshing and resilient to noise. However, as demonstrated in Figure 3 of our paper, when applied to large-scale, complex scene data (e.g., Matterport3D), CircNet tends to produce meshes with substantial holes and missing geometry. FastSESR is designed to address these scene-level challenges. While it trades off some robustness to extreme noise, it excels at generating cleaner, more complete, and higher-fidelity surfaces in large environments at high speeds.
> > >
> > > ## Robustness Contextualization against CircNet and OffsetOPT
> > >
> > > To contextualize the magnitude of performance degradation under noise, we conducted the comparative evaluation on the ABC dataset. We have included the clean data (σ = 0) baseline from our paper, alongside Gaussian noise (σ = 0.1 and σ = 0.2). The results are summarized in Table 2:
> > >
> > > Table 2: Robustness Comparison under Varying Noise Levels (ABC Dataset)
> > >
> > > | Method | Noise Level | CD1 (x10²) ↓ | CD2 (x10⁵) ↓ | F1 ↑ | NC ↑ | NR ↓ |
> > > | --- | --- | --- | --- | --- | --- | --- |
> > > | CircNet | σ = 0 (Clean) | 0.284 | 0.544 | 0.950 | 0.985 | 1.758 |
> > > | OffsetOPT | σ = 0 (Clean) | 0.283 | 0.540 | 0.951 | 0.988 | 1.318 |
> > > | FastSESR (Ours) | σ = 0 (Clean) | 0.283 | 0.540 | 0.951 | 0.988 | 1.311 |
> > > |  |  |  |  |  |  |  |
> > > | CircNet | σ = 0.1 | 0.328 | 0.720 | 0.908 | 0.965 | 10.053 |
> > > | OffsetOPT | σ = 0.1 | 5.880 | 332.094 | 0.058 | 0.522 | 55.574 |
> > > | FastSESR (Ours) | σ = 0.1 | 5.888 | 338.663 | 0.060 | 0.518 | 55.765 |
> > > |  |  |  |  |  |  |  |
> > > | CircNet | σ = 0.2 | 0.419 | 1.245 | 0.793 | 0.931 | 16.735 |
> > > | OffsetOPT | σ = 0.2 | 8.735 | 724.544 | 0.042 | 0.513 | 56.272 |
> > > | FastSESR (Ours) | σ = 0.2 | 8.727 | 733.308 | 0.043 | 0.511 | 56.339 |
> > >
> > > As the data clearly shows, while FastSESR and OffsetOPT achieve state-of-the-art accuracy on clean point clouds (σ = 0), their performance drops under severe noise is indeed significant when compared to CircNet. However, this degradation is entirely within the expected range for its specific class of methods, as evidenced by OffsetOPT's nearly identical breakdown under the same conditions. Both FastSESR and OffsetOPT rely on the local topological structure of point patches (via kNN graphs and local offset learning) to extract triangle candidates. High levels of random noise destroy these local geometric priors, causing the network to fail at identifying valid surface connections. CircNet avoids this through a completely different circumcenter-based algorithm.
> > >
> > > Impact on applications:
> > >
> > > This sensitivity indicates that FastSESR is not yet robust to severely corrupted raw scans without prior denoising or regularization. That said, our target setting is dense scene-level reconstruction, where point clouds are commonly regularized or voxelized before surface reconstruction, making such extreme noise less typical in practice. In these standard scenarios, extremely high-frequency noise is naturally filtered, allowing FastSESR to fully leverage its primary strengths: delivering state-of-the-art geometric fidelity and continuous surfaces at speeds over 20x faster than existing scene-level baselines.
> > >
> > > We will update our limitations section in the final manuscript to explicitly include this comparative context, ensuring readers have a transparent understanding of the trade-offs.

---

### Official Review · Reviewer_8LCH · 2026-03-12

**Soundness:** 3
**Presentation:** 3
**Significance:** 3
**Originality:** 3
**Overall Recommendation:** 5
**Confidence:** 5

**Summary:**

This paper presents FastSESR, a two-stage framework for scene-level explicit surface reconstruction. The method replaces the test-time optimization strategy with a feed-forward inference pipeline, and predict local triangle connectivity and an Offset Optimization Network for learning point offsets. Experiments on several scene-level datasets demonstrate that the method achieves high efficiency compared with optimization-based methods.

**Compliance With Llm Reviewing Policy:**

Affirmed.

**Final Justification:**

The authors have provided detailed clarifications and additional experiments, these additions help address my main concerns and improve the overall completeness of the evaluation. While some limitations such as sensitivity to noise and generalization still remain, they are not fundamental to the core contribution and can be further improved or clarified in future revisions. Overall, I maintain the positive assessment and raise my score.

**Key Questions For Authors:**

(1) The efficiency comparison focuses only on inference time. Reporting the training cost would provide a more complete comparison.

(2) The paper does not analyze the stability of the triangle prediction network or its robustness under different input conditions.

(3) The paper lacks discussion of limitations and representative failure cases.

**Limitations:**

Please refer to the weaknesses and key questions. I consider this is a meaningful work and would keep or raise my score if the authors address these concerns.

**Strengths And Weaknesses:**

**Strengths**

(1) This paper proposes two-stage training, which combines triangle candidate prediction with learnable offset refinement. The motivation is reasonable and allows the model to replace iterative optimization with feed-forward inference.

(2) Experiments show that the proposed method achieves substantial runtime improvements while maintaining competitive reconstruction quality on various datasets.

**Weaknesses**

(1) The efficiency comparison focuses only on inference time, while the training cost of modules such as the Triangle Candidate Network (TCN) is not reported.

(2) The paper does not analyze the stability of the triangle prediction network. Since the triangle connectivity is predicted by a pretrained model, it would be useful to evaluate its robustness under various input conditions.

(3)  Although the results show great performance on the evaluated datasets, it remains unclear in which conditions the method may failed. Therefore, the paper should add the limitation and failure cases discussion.

---

> ### Author Rebuttal · Authors · 2026-03-30
>
> Thanks for the valuable comments. We will revise the paper accordingly and add a clearer discussion of efficiency, robustness, and limitations.
>
> ## Training cost in addition to inference time
>
> All efficiency comparisons were conducted on the same 24GB NVIDIA RTX 3090, PyTorch 2.6.0 / CUDA 11.8.
> We add the training cost for stage 1:
>
> | Method | Pretrained Module | Training Cost |
> |---|---|---|
> | FastSESR(Ours) | TCN | 21 h 11 min / 100 epochs |
> | OffsetOPT(CVPR25) | TPN | 19 h 27 min / 100 epochs |
>
> Although TCN is slightly slower per 100 epochs, mainly due to additional neighborhood construction and kNN computations, it is already fit for stage-2 training after about 100 epochs. In contrast, OffsetOPT’s TPN typically requires around 300 epochs. Thus, in terms of total cost to obtain a usable pretrained model, TCN is competitive.
>
> For stage 2, OffsetOPT relies on test-time optimization, so its main cost is paid during reconstruction rather than offline training. In contrast, our approximate stage-2 training time is:
>
> | Dataset | Training Time |
> |---|---:|
> | CARLA | 10 h |
> | Matterport3D | 24.5 h |
> | ScanNet | 60 h |
>
>
> ## Stability and robustness of triangle prediction
>
> We agree that robustness should be evaluated more explicitly. We therefore added experiments under Gaussian noise and non-uniform sampling.
>
> (2.1) Noise setting
>
> We add Gaussian noise to the point cloud after normalization, following previous work CircNet (ICLR2023) and PINC (NeurIPS2023):
> $$
> x_i' = x_i + \epsilon_i,\qquad \epsilon_i \sim \mathcal{N}(0,\sigma^2 I).
> $$
> (2.2) Non-uniform setting
>
> We vary the density of the uniform data along the axis, similar to DSE (CVPR2021), to obtain a non-uniform point cloud.
>
> Results on CARLA
> | Setting               |  CD1($\times 10^{2}$)↓ |  CD2($\times 10^{5}$)↓ |   F1↑ |   NC↑ |   NR↓ |   HD↓ |
> | --------------------- | ----: | ----: | ----: | ----: | ----: | ----: |
> | FastSESR (Ours)   | 0.114 | 0.171 | 0.990 | 0.967 | 4.929 |     0.044|
> | Gaussian noise, σ=0.005 | 0.522 | 2.274 | 0.618 | 0.572 | 51.456 |0.028 |
> | Gaussian noise, σ=0.01 |   0.881 |   7.559 |   0.395 |   0.533 |   54.636 |   0.042 |
> | Non-uniform sampling  | 0.115 | 0.182 | 0.990 | 0.966 | 5.103 | 0.043 |
>
> These results suggest that FastSESR remains robust to non-uniform sampling, with only marginal degradation compared with the original input. In contrast, the performance drops noticeably under Gaussian noise, especially as the noise level increases, indicating that noise remains a clear limitation of the current method. We will present these results in the revised paper to more explicitly clarify both the robustness to non-uniform inputs and the sensitivity to noise.
>
> ## Limitations and failure cases
>
> (3.1) Our method still relies on relatively regular local neighborhoods. When the input becomes severely noisy or distribution-shifted, the triangle prior becomes less reliable, and the quality may degrade, as said in the above section.
>
> (3.2) While our paper demonstrates encouraging generalization on shape-level datasets, cross-dataset generalization on scene-level datasets is naturally more challenging, likely due to the much larger domain gap across scenes in geometry and layout. For example, the model trained on CARLA transfers less effectively to Matterport3D：
>
> | Setting               |  CD1($\times 10^{2}$)↓ |  CD2($\times 10^{5}$)↓ |   F1↑ |   NC↑ |   NR↓ |   HD↓ |
> | --------------------- | ----: | ----: | ----: | ----: | ----: | ----: |
> | CARLA->CARLA   | 0.114 | 0.171 | 0.990 | 0.967 | 4.929 |     0.044 |
> | CARLA->Matterport3D | 0.159 | 0.172 | 0.997 | 0.940 | 10.555 |0.011 |
>
> ## Reference
> Circnet: Meshing 3d point clouds with circumcenter detection.
>
> DSE: Learning delaunay surface elements for mesh reconstruction.
>
> PINC: p-Poisson surface reconstruction in curl-free flow from point clouds.

---

> > ### Author Rebuttal · Reviewer_8LCH · 2026-04-03
> >
> > Thank you for the rebuttal. The authors have provided detailed clarifications and additional experiments, these additions help address my main concerns and improve the overall completeness of the evaluation. While some limitations such as sensitivity to noise and generalization still remain, they are not fundamental to the core contribution and can be further improved or clarified in future revisions. Overall, I maintain the positive assessment and raise my score.

---

### Decision · Program_Chairs · 2026-04-30

**Decision:**

Accept (regular)

**Comment:**

3 reviewers are positive. The 4th reviewer provided negative score initially, but there is no final responses to the rebuttal. The AC carefully read all comments and agree with the positive reviewers, thus recommend clear acceptance.